# The Effects of Chlorinated Drinking Water on the Assembly of the Intestinal Microbiome

**David Martino** [1,2] 

1   Telethon Kids Institute, University of Western Australia, Perth, WA 6009, Australia;
    david.martino@telethonkids.org.au
2   Murdoch Children's Research Institute, University of Melbourne, Parkville, VIC 3052, Australia

**Abstract:** This concept paper discusses the potential impact of chlorinated public drinking water on the assembly of the intestinal microbiome in infancy. The addition of chlorine or hypochlorite to metropolitan drinking water is routinely used worldwide as a sanitizer because of its potent anti-microbial properties. It is one of the most effective means of delivering safe drinkable water because it produces a residual disinfectant that persists within the distribution system. Levels of chlorine used to treat metropolitan water are considered safe for the individual, based on toxicity studies. However, to our knowledge there have been no studies examining whether levels of persistent chlorine exposure from tap water are also safe for the ecosystem of microorganisms that colonize the gastrointestinal tract. Given the importance of the microbiome in health, persistent exposure to low levels of chlorine may be a hitherto unrecognized risk factor for gut dysbiosis, which has now been linked to virtually every chronic non-communicable disease of the modern era. Although effects may be subtle, young children and infants are more susceptible to ecological disturbance, given that the microbiome is highly influenced by environmental factors during this period. Here I outline considerations for the safety of water disinfectants not just in terms of toxicity to the host, but also for the ecosystem of microorganisms that inhabit us. Research in this is likely to bear fruitful information that could either bring attention to this issue, potentially driving new innovations in public water management; or could help confirm the safety profile of chlorine levels in public drinking water.

**Keywords:** microbial ecosystems; microbial diversity; water quality; chlorine; metagenomics

---

## 1. The Assembly of the Intestinal Microbiome in Infancy and Early Childhood

The intestinal microbiome is a complex ecosystem of microorganisms (i.e., bacteria, fungi, protozoans) that interact in a spatially and temporally structured environment within the gastrointestinal (GI) tract [1]. We now have a good appreciation of the essential duties that gut microbes play related to digestion, homeostasis of the immune system, and the production of metabolites that influence cognitive function and behavior [2]. Research shows that infants rely on colonization of the gut with commensal microorganisms to support normal development of gastrointestinal, immunological, neurological, and metabolic systems [3]. The succession of species colonizing the gut during the first three years of life is highly plastic and influenced by environmental factors, setting the scene for a stable adult microbiome [3]. The phylogenetic composition of microbial communities in the infant gut are initially dominated by *Bifidobacterium, Streptococcus, Lactococcus,* and *Lactobacillus,* and are enriched for genes involved in the de novo synthesis of folate and vitamin biosynthetic pathways throughout the first year of life [4], which generally accompanies birth and lactation. The introduction of weaning foods represents a major transition point toward a more diverse and adult-like microbiome

dominated by *Bacteriodetes*, *Prevotella*, and *Firmicutes* [5]. These trends are stereotypical and evident across multiple geographically and ethnically distinct populations [4].

The infant microbiome is initially seeded in the lower uterus and influenced by mode of birth (natural versus cesarean), use of formula milk, antibiotics, the cessation of milk feeding, and the introduction of weaning foods [5]. Cohabitation is a major influencer of enterotypes and is related to household/family practices including types of foods consumed, pets/animals in the home, and many other factors [4]. Research has shown that colonization of the GI tract has profound effects on the development and physiology of the host immune system [6], and the metabolic products of intestinal bacteria (e.g., short-chain fatty acids) influence host adaptive immune responses [7]. The assembly of the GI microbiome therefore occurs during a sensitive window during which environmental factors are more likely to cause disturbances and dysbiosis.

## 2. Dysbiosis of Intestinal Microbiota and Links to Chronic Disease

Dysbiosis of the gut microbiome causes a state of ecological imbalance when the microbial community structure loses key taxa, diversity, and/or metabolic capacity. This can lead to the bloom of opportunistic pathogens, and health consequences in early life. Microbiome research has intensified over the past decade, and evidence is rapidly accumulating that gut dysbiosis is linked to both negative childhood developmental outcomes [8] and an increased risk for chronic diseases in adulthood [1]. Given that colonization of the GI tract with microbiota is bi-directionally related to the development of the immune system, studies have naturally explored the links between dysbiosis and early immune disorders such as allergic disease. Research has uncovered links between reduced diversity of *Bifidobacterium* and *Bacteroides* and higher abundance of Enterobacteriaceae with atopic dermatitis [9]. The development of IgE-mediated food allergy in infancy and the natural resolution of food allergy in childhood have been associated with changes in the gut microbiome [10]. Young infants with an immature gut microbial composition exhibit an increased risk of asthma by age five years [11]. In addition to immune parameters, childhood obesity is associated with an altered community structure characterized by altered levels of *Firmicutes* and reduced abundance of *Bacteroides* [12]. These and other associations with chronic diseases [8] suggest dysbiosis may play an initiating role in the postnatal induction of disease risk.

## 3. The Chlorination of Public Drinking Water: A Celebrated History

One environmental factor that has the potential to disrupt the assembly of the infant microbiome, but has yet to be investigated, is chlorine levels in public drinking water. The chlorination of drinking water is a public health measure that was introduced to control microbial contamination in the early 20th century, and still remains the most common drinking water disinfectant used around the world today [13]. Controlling for microbial contamination is a major aspect of water quality management. Chlorine or hypochlorite added to metropolitan drinking water is highly toxic to microorganisms, and prevents the spread of waterborne disease. One of the earliest known uses of chlorine to disinfect public water supplies dates back to 1854 when celebrated epidemiologist Dr John Snow attempted to disinfect the Broad Street Pump water supply in London following an outbreak of cholera [14]. By the early 1900s, continuous chlorination of drinking water was successfully adopted in Great Britain, where its implementation dramatically reduced typhoid deaths. In 1908, chlorination was adopted in New Jersey in the United States and eventually extended to other towns. It virtually eliminated waterborne diseases [15], which prior to this accounted for as many deaths as modern-day road accidents. Today, more than 98% of all treated water systems in the United States employ chlorine-based disinfectants.

Although several alternative methods for disinfecting public drinking water exist, chlorine has a major advantage in that it produces a residual disinfectant that is moderately persistent [13]. It is also cheap, easy to manage, and scalable to large distribution networks. It is common practice to attempt to maintain an adequate chlorine residual throughout the distribution system. Given that chlorine exhibits potent microbicidal properties and persists in the water supply, this raises the question of

whether residual chlorine levels in the water supply act as a mild antibiotic. Indeed, enhanced antibiotic resistance for certain pathogens has controversially been associated with water chlorination, suggesting that an interaction of drinking water and gut microbiota could be a route for the dissemination of antibiotic-resistance genes [16]. Repeated and chronic exposure to low levels of chlorine may be a hitherto unrecognized risk factor in gut dysbiosis.

In Australia, the National Health and Medical Research Council infant feeding guidelines recommend boiling and cooling tap water for infant consumption until twelve months of age; this includes water used in infant formula preparation [17]. Boiling tap water removes residual chlorine, so infants are unlikely to experience significant exposure before 12 months. However, many families are unlikely to observe these guidelines and, in reality, chlorinated tap water may be introduced into the diet of many infants concurrently with weaning foods (4–6 months). While this may raise concerns, it is important to consider this in the context of the immense public health benefit water chlorination provides. The chlorine-based disinfection of raw water is essential to sustainable development, where contaminated water remains the greatest threat to public health in developing countries. In 1990, diarrheal disease caused by waterborne pathogens killed more than three million young children under the age of five [14]. Estimates suggest that the disinfection of public drinking water has contributed to a 50% increase in life expectancy in developed countries in the 20th century [14], chiefly through the control of waterborne diseases such as cholera, typhoid, dysentery, and hepatitis A. Clearly, water chlorination protects against a substantial burden of childhood mortality, and this message must be balanced against any assertions related to a theoretical increase in childhood morbidity.

## 4. The Safety Profile of Chlorine

Public drinking water is safe for people of all ages, including children over six months of age and the very old. In Australia, the National standard mandates that levels of chlorine in the water scheme should not exceed 5 mg/L [13], which is consistent with guideline values recommended by the World Health Organization (WHO) [18]. These guidelines were established in 1993 as a conservative value based on extensive animal testing and human observational studies. Animal studies in numerous different rodent models over short and long exposure periods and over a range of doses suggest that the chlorine levels in public drinking water are safe. The safety profile of chlorine has been examined using end-points of animal physiology, carcinogenicity, reproductive effects, teratogenicity, and embryotoxicity [18].

There have been more recent concerns around the world surrounding chlorination by-products created during water disinfection. Chlorine reacts with certain natural organic material in water supplies to form trihalomethanes (THMs) including chloroform, bromodichloromethane, dibromochloromethane, and bromoform [19]. These by-products have been found to be fairly ubiquitous and can reach levels as high as 160 μg/L [19]. Since the discovery of THMs in the 1970s, epidemiological studies have suggested associations between the exposure to elevated levels of THMs and bladder cancer [20], miscarriage [21], and babies being born small for their gestational age [22]. Overall, these studies suggest small positive effects on disease risk, and that the magnitude of absolute risk is difficult to ascertain as most studies report odds ratios rather than relative risks. Studies are associative in nature and subject to confounding, although the biological plausibility for potential toxic effects of THMs is clear, as these compounds are easily absorbed through the skin, lungs, and gut. After extensive expert evaluation of every major scientific evaluation, the International Agency for Research on Cancer (IARC) has concluded that there is not enough evidence to prove that THMs pose a health risk, yet research is ongoing.

Due to concerns surrounding chlorination by-products, utilities are increasingly exploring a variety of alternative strategies toward water disinfection, including combinations of primary disinfectants (e.g., UV, ozone) with chloramines as secondary disinfectants to reduce THMs [23]. Maintaining the fine balance between the acute risk associated with pathogen control and the chronic risk arising from lifetime exposure to disinfection by-products is a major challenge. The benefits of changing the type of disinfectant come at the expense of the enhanced production of other by-products

such as bromates, halonitromethanes, and haloacetaldehydes [19], as has been observed in parts of Europe [24]. Fears that THMs could be a potential carcinogen prompted the US Environmental Protection Agency (EPA) to set regulatory limits for disinfection by-products. The resulting "chemophobia" led to outbreaks of cholera in Peru in 1991 due to inadequate water disinfection due in part to US concerns over the potential risks from disinfection by-products. The outbreak spread to 19 Latin American countries, causing a million cases and 10,000 reported deaths. These examples illustrate the potentially disastrous effects of public perception and lackadaisical water disinfection measures. Despite this, chlorine-free drinking water has been achieved in the Netherlands due to the availability of high-quality source water and the use of biostable distribution materials with rigorous monitoring [25], which may serve as a model for future developments in water treatment practices.

While the evidence suggests the levels of residual chlorine in the water scheme are safe, this has only been measured in the context of toxicity to the individual, and we have not yet considered that it may be toxic to the billions of microorganisms that colonize the GI tract. No studies have yet investigated this in people at the time of writing.

## 5. How Might Ingested Chlorine Affect the Intestinal Microbiota?

One issue with the hypothesis articulated above is that it is unclear whether ingested chlorine residuals would passage through the stomach and contact the commensal microorganisms that reside in the intestines. It seems unlikely, given that stomach acid itself is highly chlorinated and composed of hydrochloric acid, sodium, and potassium chloride. Perhaps a more likely scenario would be THM by-products being absorbed in the bowel and disrupting the microbiota. Evidence from mouse studies suggest that ingested disinfection by-products were associated with changes in fecal microbial diversity characterized by an elevated relative abundance of *Bacteroidetes* and dose-dependent changes in the ratio of *Firmicutes/Bacteroidetes* [26]. In one mouse study, prolonged exposure to high concentrations of chlorinated tap water (i.e., 10 mg/L, twice the WHO standard for humans) reduced the number of *C. perfringens*, *C. difficile*, *Enterobacteriaceae*, and *Staphylococcus* counts in fecal samples [27]. Other mouse studies have demonstrated that exposure to tap water induces changes in clinically relevant taxa in fecal samples [28]. This may occur either via the effects of chlorine, chlorine by-products, or increased bacterial exposure from home water filters [29]. A recent metagenomic analysis of fecal samples from 60 healthy twins sampled from 0–8 months of life reported associations between domestic water sources and associated microbial signatures [30]. Tap water exposure predicted a decrease in *Enterobacteriaceae*, glycogen degradation pathways, and homolactic fermentation. Together with the animal data, these patterns suggest that drinking water quality influences the acquisition of the microbiome.

## 6. How Might the Effects of Chlorinated Drinking Water on the Microbiome Be Studied?

Whilst experiments to study the effects of chlorinated water on the microbiome have been conducted in mice [26–28] it is unclear how generalizable the findings from these studies are. Observational human studies examining changes in fecal microbiota composition in relation to water quality have recently emerged, and provide the first suggestive associations [30]. A randomized intervention trial carried out in a single geographical area could be an ideal approach to investigating the potential effects discussed here. A representative sample of families birthing at a local health campus could be recruited and randomized to two groups to experience the following drinking water regimes: (1) Unmodified tap water as distributed by the local public water supply; (2) Filtered tap water by means of activated carbon ion exchange to remove dissolved chlorine and other contaminants. A benchtop water filtration unit fitted with a 0.5 μm carbon cartridge that effectively removes chloramines, chlorine, pesticides, and heavy metals could be installed in homes. Sham filters could be provided to the control group, and water quality measures for all families could be ascertained directly by sampling the water from the tap by chemical analysis, and/or through accessing data on water quality through local municipalities. The ideal intervention period would start prior to any exposure to chlorinated water—for example, 6 months of age when infants are predominantly breast-

or formula-fed. Sampling of the stool microbiome through metagenomic sequencing could occur at 6 months to provide a baseline. As children get older, they will gradually be exposed to un-boiled tap water as a regular part of their diet. Re-sampling the stool microbiome could then facilitate a follow-up sample after a sufficient period of intervention (e.g., 1 year). The primary outcome would be changes in the microbiome. Between-intervention effects on richness, diversity indices, and relative abundance of individual operational taxonomic unities (OTUs), genera, and species should be tested using linear mixed models, with a random effect of the subject. Between-intervention effects on community structure could be assessed by principal coordinate ordination. By inclusion of direct chemical analysis of water samples from family homes, one could use regression analyses to identify specific enteric microbial species that are influenced by water quality parameters. Of course, extensive lifestyle and environmental data would be needed to control for confounders. A study such as this might provide foundational evidence for an effect of water chlorination on the intestinal microbiome, and could potentially justify further studies including early life health outcomes such as allergy and asthma in children.

## 7. Concluding Remarks

The transition to modernity has brought about significant benefits to human populations at the cost of ecological disturbance. Ecological changes brought about by sanitation and improved public health are associated with rising rates of chronic non-communicable diseases in many parts of the world. This concept paper proposes a theoretical role for residual chlorine levels arising from water disinfection practices as a potential risk factor for intestinal dysbiosis. Preliminary evidence from animal studies suggests that chlorinated water might impact the microbiome directly or via secondary disinfection by-products. The challenges of studying this in human populations are substantial. Chlorine levels in tap water vary throughout the day depending on tap use, and are likely to vary between households, depending on distance of the home to a distribution point, and between municipalities. Practices around boiling water for infants and children are also likely to vary substantially across households. Adherence to guidelines that recommend boiling water for the first 12 months are likely to determine exposure levels, and research in this area will help determine whether these guidelines are appropriate to protect the developing microbiome. The challenge will be to detect these subtle effects against a background of developmental changes in community structure and organization that accompany child development. Negative or positive findings in this area will be highly informative and likely of broad interest to individuals, local and state governments, as well as the academic community.

**Funding:** This research was supported by the National Health and Medical Research Council of Australia through their fellowship scheme.

**Acknowledgments:** I would like to thank the WA State Department of Health and the Water Corporation of Western Australia for discussions on this topic.

**Conflicts of Interest:** The author declares no conflict of interest.

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
