# Peer review of "The Effects of Chlorinated Drinking Water on the Assembly of the Intestinal Microbiome"

_challenges, doi:10.3390/challe10010010_

Round 1

Reviewer 1 Report

The proposed concept paper discusses the role played by chlorinated public drinking water on the assembly of intestinal microbiome in infancy.

The paper is clearly written, but it could be improved including more references and deepening some important issues. For this reason, it could be suitable for publication after the following major revisions.

1) The Section 1 does not provide sufficient references and different sentences are without a reference (see lines 30-34; 43-47). 

2) Section 3, lines 97-98 "In Australia the pediatric guidelines recommend boiling tap water for the first twelve months of life, this includes water used in infant formula preparation". Which guidelines? Could the Author give a better explanation of the method proposed?

3) Section 4. This section must be improved by the Author. In particular, it could be of interest to better define the risk due to THMs exposures in causing health damages. It could be also interesting to describe the possible role played by THMs in the developing of oncological diseases also involving the intestine, and what happen when the disinfectant is changed. A description to the world panorama shoul be given. For example, in Italy the hypochlorite is almost no more used during the final disinfection step, and the THMs presence in drinking water is decreased. But the changing of the disinfectant could led to other by-products appearence, which could affect people and infant (see for example Azara et al, "Derogation from drinking water quality standards in Italy according to the European Directive 98/83/EC and the Legislative Decree 31/2001 - a look at the recent past; Azara et al, "First results on the use of chloramines to reduce disinfection byproducts in drinking water"; Dettori et al, doi:107416.ai.2016.2109). All these issues must be included in the discussion and a focus on the situation in other Countries must be included too.

Author Response

Thanks to the reviewer for their helpful comments on the manuscript. I've addressed the suggestions below:

1) The Section 1 does not provide sufficient references and different sentences are without a reference (see lines 30-34; 43-47). 

Response: In the revision Section 1 is now more carefully referenced, specifically at the points mentioned

2) Section 3, lines 97-98 "In Australia the pediatric guidelines recommend boiling tap water for the first twelve months of life, this includes water used in infant formula preparation". Which guidelines? Could the Author give a better explanation of the method proposed?

Response: In the revision on lines 102-104 I've now indicated that these guidelines are the National Health and Medical Research Council guidelines on Infant Feeding with reference provided. 

"     In Australia the National Health and Medical Research Council infant feeding guidelines recommend boiling and cooling tap water for infant consumption until twelve months of age, this includes water used in infant formula preparation [ref included]. "

3) Section 4. This section must be improved by the Author. In particular, it could be of interest to better define the risk due to THMs exposures in causing health damages. It could be also interesting to describe the possible role played by THMs in the developing of oncological diseases also involving the intestine, and what happen when the disinfectant is changed. A description to the world panorama shoul be given. For example, in Italy the hypochlorite is almost no more used during the final disinfection step, and the THMs presence in drinking water is decreased. But the changing of the disinfectant could led to other by-products appearence, which could affect people and infant (see for example Azara et al, "Derogation from drinking water quality standards in Italy according to the European Directive 98/83/EC and the Legislative Decree 31/2001 - a look at the recent past; Azara et al, "First results on the use of chloramines to reduce disinfection byproducts in drinking water"; Dettori et al, doi:107416.ai.2016.2109). All these issues must be included in the discussion and a focus on the situation in other Countries must be included too.

Response: I appreciate this point and have now included a more thorough discussion of THMs covering the topics suggested by the reviewer in Section 4 (L119-164) although I've kept it brief as this paper is intended as a commentary with a specific focus on the gut microbiome and is not intended as a thorough review. 

Reviewer 2 Report

This manuscript proposed a study on the potential impact of chlorinated public drinking water on  the intestinal microbiome ecosystem in infancy. The topic is interesting while challenging.

The author raised the question but failed to provide the solution to the question. It would be better to discuss potential methods/approaches as well as workflows to carry out such a study.

Author Response

Thanks to the reviewer for the helpful suggestions. A response to comments is provided below:

1) This manuscript proposed a study on the potential impact of chlorinated public drinking water on  the intestinal microbiome ecosystem in infancy. The topic is interesting while challenging.

The author raised the question but failed to provide the solution to the question. It would be better to discuss potential methods/approaches as well as workflows to carry out such a study.

Response: I don't believe this paper proposed a study anywhere in the original text, it was merely intended to discuss the concept. However, in line with the reviewers advice I've now included a new section 6 (L186-211) outlining a potential approach to addressing this complex issue. Copied here for convenience:

6. How might the effects of chlorinated drinking water on the microbiome be studied?

            A randomized intervention trial carried out in a single geographical area could be an ideal approach to investigating the potential effects discussed here. A representative sample of families birthing at a local health campus could be recruited and randomized to two groups to experience the following drinking water regimes: [1] Unmodified tap water as distributed by the local public water supply [2] Filtered tap water by means of activated carbon ion exchange to remove dissolved chlorine and other contaminants. A benchtop water filtration unit fitted with a 0.5 micron carbon cartridge could be installed in home that effectively removes chloramines, chlorine, pesticides and heavy metals. Sham filters could be provided to the control group and water quality measures for all families could be ascertained directly by sampling the water from the tap by chemical analysis, and/or through accessing data on water quality through local municipalities. The ideal intervention period would start prior to any exposure to chlorinated water, for example, 6 months of age when infants are predominantly breast or formula fed. Sampling of the stool microbiome through metagenomic sequencing could occur 6 months to provide a baseline. As children get older, they will gradually be exposed to un-boiled tap water as a regular part of their diet. Re-sampling the stool microbiome could then facilitate a follow up sample after a sufficient period of intervention, for example 1 year. The primary outcome would be changes in the microbiome. Between-intervention effect on richness, diversity indices and relative abundance of individual operational taxonomic unities (OTUs), genera and species should be tested using linear mixed models, with a random effect of subject. Between-intervention effect on community structure can be assessed by principal coordinate ordination. By inclusion of direct chemical analysis of water samples from family homes one could use regression analyses to identify specific enteric microbial species that are influenced by water quality parameters. Of course, extensive lifestyle and environmental data would be needed to control for confounding. A study such as this might provide foundational evidence for an effect of water chlorination on the intestinal microbiome and could potentially justify further studies including early life health outcomes such as allergy and asthma in children.  

Round 2

Reviewer 1 Report

All the issues raised have been addressed by the Author.

Reviewer 2 Report

The author addressed my questions. No more comments.